# Accelerating Stochastic Gradient Descent using Predictive Variance Reduction

**Rie Johnson**
RJ Research Consulting
Tarrytown NY, USA

**Tong Zhang**
Baidu Inc., Beijing, China
Rutgers University, New Jersey, USA

## Abstract

Stochastic gradient descent is popular for large scale optimization but has slow convergence asymptotically due to the inherent variance. To remedy this problem, we introduce an explicit variance reduction method for stochastic gradient descent which we call stochastic variance reduced gradient (SVRG). For smooth and strongly convex functions, we prove that this method enjoys the same fast convergence rate as those of stochastic dual coordinate ascent (SDCA) and Stochastic Average Gradient (SAG). However, our analysis is significantly simpler and more intuitive. Moreover, unlike SDCA or SAG, our method does not require the storage of gradients, and thus is more easily applicable to complex problems such as some structured prediction problems and neural network learning.

## 1 Introduction

In machine learning, we often encounter the following optimization problem. Let $\psi_1, \ldots, \psi_n$ be a sequence of vector functions from $\mathbb{R}^d$ to $\mathbb{R}$. Our goal is to find an approximate solution of the following optimization problem

$$\min P(w), \qquad P(w) := \frac{1}{n} \sum_{i=1}^{n} \psi_i(w). \tag{1}$$

For example, given a sequence of $n$ training examples $(x_1, y_1), \ldots, (x_n, y_n)$, where $x_i \in \mathbb{R}^d$ and $y_i \in \mathbb{R}$, if we use the squared loss $\psi_i(w) = (w^\top x_i - y_i)^2$, then we can obtain least squares regression. In this case, $\psi_i(\cdot)$ represents a loss function in machine learning. One may also include regularization conditions. For example, if we take $\psi_i(w) = \ln(1 + \exp(-w^\top x_i y_i)) + 0.5\lambda w^\top w$ ($y_i \in \{\pm 1\}$), then the optimization problem becomes regularized logistic regression.

A standard method is gradient descent, which can be described by the following update rule for $t = 1, 2, \ldots$

$$w^{(t)} = w^{(t-1)} - \eta_t \nabla P(w^{(t-1)}) = w^{(t-1)} - \frac{\eta_t}{n} \sum_{i=1}^{n} \nabla \psi_i(w^{(t-1)}). \tag{2}$$

However, at each step, gradient descent requires evaluation of $n$ derivatives, which is expensive. A popular modification is stochastic gradient descent (SGD): where at each iteration $t = 1, 2, \ldots$, we draw $i_t$ randomly from $\{1, \ldots, n\}$, and

$$w^{(t)} = w^{(t-1)} - \eta_t \nabla \psi_{i_t}(w^{(t-1)}). \tag{3}$$

The expectation $\mathbb{E}[w^{(t)}|w^{(t-1)}]$ is identical to (2). A more general version of SGD is the following

$$w^{(t)} = w^{(t-1)} - \eta_t g_t(w^{(t-1)}, \xi_t), \tag{4}$$

where $\xi_t$ is a random variable that may depend on $w^{(t-1)}$, and the expectation (with respect to $\xi_t$) $\mathbb{E}[g_t(w^{(t-1)}, \xi_t)|w^{(t-1)}] = \nabla P(w^{(t-1)})$. The advantage of stochastic gradient is that each step only relies on a single derivative $\nabla\psi_i(\cdot)$, and thus the computational cost is $1/n$ that of the standard gradient descent. However, a disadvantage of the method is that the randomness introduces variance — this is caused by the fact that $g_t(w^{(t-1)}, \xi_t)$ equals the gradient $\nabla P(w^{(t-1)})$ in expectation, but each $g_t(w^{(t-1)}, \xi_t)$ is different. In particular, if $g_t(w^{(t-1)}, \xi_t)$ is large, then we have a relatively large variance which slows down the convergence. For example, consider the case that each $\psi_i(w)$ is smooth

$$\psi_i(w) - \psi_i(w') - 0.5L\|w - w'\|_2 \leq \nabla\psi_i(w')^\top (w - w'), \tag{5}$$

and convex; and $P(w)$ is strongly convex

$$P(w) - P(w') - 0.5\gamma\|w - w'\|_2^2 \geq \nabla P(w')^\top (w - w'), \tag{6}$$

where $L \geq \gamma \geq 0$. As long as we pick $\eta_t$ as a constant $\eta < 1/L$, we have linear convergence of $O((1 - \gamma/L)^t)$ Nesterov [2004]. However, for SGD, due to the variance of random sampling, we generally need to choose $\eta_t = O(1/t)$ and obtain a slower sub-linear convergence rate of $O(1/t)$. This means that we have a trade-off of fast computation per iteration and slow convergence for SGD versus slow computation per iteration and fast convergence for gradient descent. Although the fast computation means it can reach an approximate solution relatively quickly, and thus has been proposed by various researchers for large scale problems Zhang [2004], Shalev-Shwartz et al. [2007] (also see Leon Bottou's Webpage `http://leon.bottou.org/projects/sgd`), the convergence slows down when we need a more accurate solution.

In order to improve SGD, one has to design methods that can reduce the variance, which allows us to use a larger learning rate $\eta_t$. Two recent papers Le Roux et al. [2012], Shalev-Shwartz and Zhang [2012] proposed methods that achieve such a variance reduction effect for SGD, which leads to a linear convergence rate when $\psi_i(w)$ is smooth and strongly convex. The method in Le Roux et al. [2012] was referred to as SAG (stochastic average gradient), and the method in Shalev-Shwartz and Zhang [2012] was referred to as SDCA. These methods are suitable for training convex linear prediction problems such as logistic regression or least squares regression, and in fact, SDCA is the method implemented in the popular lib-SVM package Hsieh et al. [2008]. However, both proposals require storage of all gradients (or dual variables). Although this issue may not be a problem for training simple regularized linear prediction problems such as least squares regression, the requirement makes it unsuitable for more complex applications where storing all gradients would be impractical. One example is training certain structured learning problems with convex loss, and another example is training nonconvex neural networks. In order to remedy the problem, we propose a different method in this paper that employs explicit variance reduction without the need to store the intermediate gradients. We show that if $\psi_i(w)$ is strongly convex and smooth, then the same convergence rate as those of Le Roux et al. [2012], Shalev-Shwartz and Zhang [2012] can be obtained. Even if $\psi_i(w)$ is nonconvex (such as neural networks), under mild assumptions, it can be shown that asymptotically the variance of SGD goes to zero, and thus faster convergence can be achieved. In summary, this work makes the following three contributions:

- Our method does not require the storage of full gradients, and thus is suitable for some problems where methods such as Le Roux et al. [2012], Shalev-Shwartz and Zhang [2012] cannot be applied.

- We provide a much simpler proof of the linear convergence results for smooth and strongly convex loss, and our view provides a significantly more intuitive explanation of the fast convergence by explicitly connecting the idea to variance reduction in SGD. The resulting insight can easily lead to additional algorithmic development.

- The relatively intuitive variance reduction explanation also applies to nonconvex optimization problems, and thus this idea can be used for complex problems such as training deep neural networks.

## 2  Stochastic Variance Reduced Gradient

One practical issue for SGD is that in order to ensure convergence the learning rate $\eta_t$ has to decay to zero. This leads to slower convergence. The need for a small learning rate is due to the variance

of SGD (that is, SGD approximates the full gradient using a small batch of samples or even a single example, and this introduces variance). However, there is a fix described below. At each time, we keep a version of estimated $w$ as $\tilde{w}$ that is close to the optimal $w$. For example, we can keep a snapshot of $\tilde{w}$ after every $m$ SGD iterations. Moreover, we maintain the average gradient

$$\tilde{\mu} = \nabla P(\tilde{w}) = \frac{1}{n} \sum_{i=1}^{n} \nabla \psi_i(\tilde{w}),$$

and its computation requires one pass over the data using $\tilde{w}$. Note that the expectation of $\nabla \psi_i(\tilde{w}) - \tilde{\mu}$ over $i$ is zero, and thus the following update rule is generalized SGD: randomly draw $i_t$ from $\{1, \ldots, n\}$:

$$w^{(t)} = w^{(t-1)} - \eta_t(\nabla \psi_i(w^{(t-1)}) - \nabla \psi_{i_t}(\tilde{w}) + \tilde{\mu}). \tag{7}$$

We thus have

$$\mathbb{E}[w^{(t)}|w^{(t-1)}] = w^{(t-1)} - \eta_t \nabla P(w^{(t-1)}).$$

That is, if we let the random variable $\xi_t = i_t$ and $g_t(w^{(t-1)}, \xi_t) = \nabla \psi_{i_t}(w^{(t-1)}) - \nabla \psi_{i_t}(\tilde{w}) + \tilde{\mu}$, then (7) is a special case of (4).

The update rule in (7) can also be obtained by defining the auxiliary function

$$\tilde{\psi}_i(w) = \psi_i(w) - (\nabla \psi_i(\tilde{w}) - \tilde{\mu})^\top w.$$

Since $\sum_{i=1}^{n} (\nabla \psi_i(\tilde{w}) - \tilde{\mu}) = 0$, we know that

$$P(w) = \frac{1}{n} \sum_{i=1}^{n} \psi_i(w) = \frac{1}{n} \sum_{i=1}^{n} \tilde{\psi}_i(w).$$

Now we may apply the standard SGD to the new representation $P(w) = \frac{1}{n} \sum_{i=1}^{n} \tilde{\psi}_i(w)$ and obtain the update rule (7).

To see that the variance of the update rule (7) is reduced, we note that when both $\tilde{w}$ and $w^{(t)}$ converge to the same parameter $w_*$, then $\tilde{\mu} \to 0$. Therefore if $\nabla \psi_i(\tilde{w}) \to \nabla \psi_i(w_*)$, then

$$\nabla \psi_i(w^{(t-1)}) - \nabla \psi_i(\tilde{w}) + \tilde{\mu} \to \nabla \psi_i(w^{(t-1)}) - \nabla \psi_i(w_*) \to 0.$$

This argument will be made more rigorous in the next section, where we will analyze the algorithm in Figure 1 that summarizes the ideas described in this section. We call this method *stochastic variance reduced gradient (SVRG)* because it explicitly reduces the variance of SGD. Unlike SGD, the learning rate $\eta_t$ for SVRG does not have to decay, which leads to faster convergence as one can use a relatively large learning rate. This is confirmed by our experiments.

In practical implementations, it is natural to choose option I, or take $\tilde{w}_s$ to be the average of the past $t$ iterates. However, our analysis depends on option II. Note that each stage $s$ requires $2m + n$ gradient computations (for some convex problems, one may save the intermediate gradients $\nabla \psi_i(\tilde{w})$, and thus only $m + n$ gradient computations are needed). Therefore it is natural to choose $m$ to be the same order of $n$ but slightly larger (for example $m = 2n$ for convex problems and $m = 5n$ for nonconvex problems in our experiments). In comparison, standard SGD requires only $m$ gradient computations. Since gradient may be the computationally most intensive operation, for fair comparison, we compare SGD to SVRG based on the number of gradient computations.

## 3 Analysis

For simplicity we will only consider the case that each $\psi_i(w)$ is convex and smooth, and $P(w)$ is strongly convex.

**Theorem 1.** *Consider SVRG in Figure 1 with option II. Assume that all $\psi_i$ are convex and both (5) and (6) hold with $\gamma > 0$. Let $w_* = \arg\min_w P(w)$. Assume that $m$ is sufficiently large so that*

$$\alpha = \frac{1}{\gamma\eta(1 - 2L\eta)m} + \frac{2L\eta}{1 - 2L\eta} < 1,$$

*then we have geometric convergence in expectation for SVRG:*

$$\mathbb{E}\, P(\tilde{w}_s) \leq \mathbb{E}\, P(w_*) + \alpha^s [P(\tilde{w}_0) - P(w_*)]$$

<div style="border:1px solid black; padding:10px;">

**Procedure SVRG**

**Parameters** update frequency $m$ and learning rate $\eta$
**Initialize** $\tilde{w}_0$
**Iterate:** for $s = 1, 2, \ldots$
  $\tilde{w} = \tilde{w}_{s-1}$
  $\tilde{\mu} = \frac{1}{n}\sum_{i=1}^n \nabla\psi_i(\tilde{w})$
  $w_0 = \tilde{w}$
  **Iterate:** for $t = 1, 2, \ldots, m$
    Randomly pick $i_t \in \{1, \ldots, n\}$ and update weight
      $w_t = w_{t-1} - \eta(\nabla\psi_{i_t}(w_{t-1}) - \nabla\psi_{i_t}(\tilde{w}) + \tilde{\mu})$
  **end**
  **option I**: set $\tilde{w}_s = w_m$
  **option II**: set $\tilde{w}_s = w_t$ for randomly chosen $t \in \{0, \ldots, m-1\}$
**end**

</div>

Figure 1: Stochastic Variance Reduced Gradient

*Proof.* Given any $i$, consider

$$g_i(w) = \psi_i(w) - \psi_i(w_*) - \nabla\psi_i(w_*)^\top(w - w_*).$$

We know that $g_i(w_*) = \min_w g_i(w)$ since $\nabla g_i(w_*) = 0$. Therefore

$$0 = g_i(w_*) \leq \min_\eta[g_i(w - \eta\nabla g_i(w))]$$

$$\leq \min_\eta[g_i(w) - \eta\|\nabla g_i(w)\|_2^2 + 0.5L\eta^2\|\nabla g_i(w)\|_2^2] = g_i(w) - \frac{1}{2L}\|\nabla g_i(w)\|_2^2.$$

That is,

$$\|\nabla\psi_i(w) - \nabla\psi_i(w_*)\|_2^2 \leq 2L[\psi_i(w) - \psi_i(w_*) - \nabla\psi_i(w_*)^\top(w - w_*)].$$

By summing the above inequality over $i = 1, \ldots, n$, and using the fact that $\nabla P(w_*) = 0$, we obtain

$$n^{-1}\sum_{i=1}^n \|\nabla\psi_i(w) - \nabla\psi_i(w_*)\|_2^2 \leq 2L[P(w) - P(w_*)]. \tag{8}$$

We can now proceed to prove the theorem. Let $v_t = \nabla\psi_{i_t}(w_{t-1}) - \nabla\psi_{i_t}(\tilde{w}) + \tilde{\mu}$. Conditioned on $w_{t-1}$, we can take expectation with respect to $i_t$, and obtain:

$$\mathbb{E}\,\|v_t\|_2^2$$
$$\leq 2\,\mathbb{E}\,\|\nabla\psi_{i_t}(w_{t-1}) - \nabla\psi_{i_t}(w_*)\|_2^2 + 2\,\mathbb{E}\,\|[\nabla\psi_{i_t}(\tilde{w}) - \nabla\psi_{i_t}(w_*)] - \nabla P(\tilde{w})\|_2^2$$
$$= 2\,\mathbb{E}\,\|\nabla\psi_{i_t}(w_{t-1}) - \nabla\psi_{i_t}(w_*)\|_2^2 + 2\,\mathbb{E}\,\|[\nabla\psi_{i_t}(\tilde{w}) - \nabla\psi_{i_t}(w_*)]$$
$$\qquad - \mathbb{E}[\nabla\psi_{i_t}(\tilde{w}) - \nabla\psi_{i_t}(w_*)]\|_2^2$$
$$\leq 2\,\mathbb{E}\,\|\nabla\psi_{i_t}(w_{t-1}) - \nabla\psi_{i_t}(w_*)\|_2^2 + 2\,\mathbb{E}\,\|\nabla\psi_{i_t}(\tilde{w}) - \nabla\psi_{i_t}(w_*)\|_2^2$$
$$\leq 4L[P(w_{t-1}) - P(w_*) + P(\tilde{w}) - P(w_*)].$$

The first inequality uses $\|a + b\|_2^2 \leq 2\|a\|_2^2 + 2\|b\|_2^2$ and $\tilde{\mu} = \nabla P(\tilde{w})$. The second inequality uses $\mathbb{E}\,\|\xi - \mathbb{E}\,\xi\|_2^2 = \mathbb{E}\,\|\xi\|_2^2 - \|\mathbb{E}\,\xi\|_2^2 \leq \mathbb{E}\,\|\xi\|_2^2$ for any random vector $\xi$. The third inequality uses (8).

Now by noticing that conditioned on $w_{t-1}$, we have $\mathbb{E}\,v_t = \nabla P(w_{t-1})$; and this leads to

$$\mathbb{E}\,\|w_t - w_*\|_2^2$$
$$= \|w_{t-1} - w_*\|_2^2 - 2\eta(w_{t-1} - w_*)^\top\,\mathbb{E}\,v_t + \eta^2\,\mathbb{E}\,\|v_t\|_2^2$$
$$\leq \|w_{t-1} - w_*\|_2^2 - 2\eta(w_{t-1} - w_*)^\top\nabla P(w_{t-1}) + 4L\eta^2[P(w_{t-1}) - P(w_*) + P(\tilde{w}) - P(w_*)]$$
$$\leq \|w_{t-1} - w_*\|_2^2 - 2\eta[P(w_{t-1}) - P(w_*)] + 4L\eta^2[P(w_{t-1}) - P(w_*) + P(\tilde{w}) - P(w_*)]$$
$$= \|w_{t-1} - w_*\|_2^2 - 2\eta(1 - 2L\eta)[P(w_{t-1}) - P(w_*)] + 4L\eta^2[P(\tilde{w}) - P(w_*)].$$

The first inequality uses the previously obtained inequality for $\mathbb{E}\|v_t\|_2^2$, and the second inequality convexity of $P(w)$, which implies that $-(w_{t-1} - w_*)^\top \nabla P(w_{t-1}) \leq P(w_*) - P(w_{t-1})$.

We consider a fixed stage s, so that $\tilde{w} = \tilde{w}_{s-1}$ and $\tilde{w}_s$ is selected after all of the updates have completed. By summing the previous inequality over $t = 1, \ldots, m$, taking expectation with all the history, and using option II at stage $s$, we obtain

$$\mathbb{E}\,\|w_m - w_*\|_2^2 + 2\eta(1 - 2L\eta)m\,\mathbb{E}\,[P(\tilde{w}_s) - P(w_*)]$$

$$\leq \mathbb{E}\,\|w_0 - w_*\|_2^2 + 4Lm\eta^2\,\mathbb{E}[P(\tilde{w}) - P(w_*)]$$

$$= \mathbb{E}\,\|\tilde{w} - w_*\|_2^2 + 4Lm\eta^2\,\mathbb{E}[P(\tilde{w}) - P(w_*)]$$

$$\leq \frac{2}{\gamma}\,\mathbb{E}[P(\tilde{w}) - P(w_*)] + 4Lm\eta^2\,\mathbb{E}[P(\tilde{w}) - P(w_*)]$$

$$= 2(\gamma^{-1} + 2Lm\eta^2)\,\mathbb{E}[P(\tilde{w}) - P(w_*)].$$

The second inequality uses the strong convexity property (6). We thus obtain

$$\mathbb{E}\,[P(\tilde{w}_s) - P(w_*)] \leq \left[ \frac{1}{\gamma\eta(1 - 2L\eta)m} + \frac{2L\eta}{1 - 2L\eta} \right] \mathbb{E}[P(\tilde{w}_{s-1}) - P(w_*)].$$

This implies that $\mathbb{E}\,[P(\tilde{w}_s) - P(w_*)] \leq \alpha^s\,\mathbb{E}\,[P(\tilde{w}_0) - P(w_*)]$. The desired bound follows. □

The bound we obtained in Theorem 1 is comparable to those obtained in Le Roux et al. [2012], Shalev-Shwartz and Zhang [2012] (if we ignore the log factor). To see this, we may consider for simplicity the most indicative case where the condition number $L/\gamma = n$. Due to the poor condition number, the standard batch gradient descent requires complexity of $n \ln(1/\epsilon)$ iterations over the data to achieve accuracy of $\epsilon$, which means we have to process $n^2 \ln(1/\epsilon)$ number of examples. In comparison, in our procedure we may take $\eta = 0.1/L$ and $m = O(n)$ to obtain a convergence rate of $\alpha = 1/2$. Therefore to achieve an accuracy of $\epsilon$, we need to process $n \ln(1/\epsilon)$ number of examples. This matches the results of Le Roux et al. [2012], Shalev-Shwartz and Zhang [2012]. Nevertheless, our analysis is significantly simpler than both Le Roux et al. [2012] and Shalev-Shwartz and Zhang [2012], and the explicit variance reduction argument provides better intuition on why this method works. In fact, in Section 4 we show that a similar intuition can be used to explain the effectiveness of SDCA.

The SVRG algorithm can also be applied to smooth but not strongly convex problems. A convergence rate of $O(1/T)$ may be obtained, which improves the standard SGD convergence rate of $O(1/\sqrt{T})$. In order to apply SVRG to nonconvex problems such as neural networks, it is useful to start with an initial vector $\tilde{w}_0$ that is close to a local minimum (which may be obtained with SGD), and then the method can be used to accelerate the local convergence rate of SGD (which may converge very slowly by itself). If the system is locally (strongly) convex, then Theorem 1 can be directly applied, which implies local geometric convergence rate with a constant learning rate.

## 4 SDCA as Variance Reduction

It can be shown that both SDCA and SAG are connected to SVRG in the sense they are also a variance reduction methods for SGD, although using different techniques. In the following we present the variance reduction view of SDCA, which provides additional insights into these recently proposed fast convergence methods for stochastic optimization. In SDCA, we consider the following problem with convex $\phi_i(w)$:

$$w_* = \arg\min P(w), \qquad P(w) = \frac{1}{n}\sum_{i=1}^{n}\phi_i(w) + 0.5\lambda w^\top w. \qquad (9)$$

This is the same as our formulation with $\psi_i(w) = \phi_i(w) + 0.5\lambda w^\top w$.

We can take the derivative of (9) and derive a "dual" representation of $w$ at the solution $w_*$ as:

$$w_* = \sum_{i=1}^{n}\alpha_i^* \qquad (j = 1, \ldots, k),$$

where the dual variables

$$\alpha_i^* = -\frac{1}{\lambda n}\nabla\phi_i(w_*). \tag{10}$$

Therefore in the SGD update (3), if we maintain a representation

$$w^{(t)} = \sum_{i=1}^{n}\alpha_i^{(t)}, \tag{11}$$

then the update of $\alpha$ becomes:

$$\alpha_\ell^{(t)} = \begin{cases} (1-\eta_t\lambda)\alpha_i^{(t-1)} - \eta_t\nabla\phi_i(w) & \ell = i \\ (1-\eta_t\lambda)\alpha_\ell^{(t-1)} & \ell \neq i \end{cases}. \tag{12}$$

This update rule requires $\eta_t \to 0$ when $t \to \infty$.

Alternatively, we may consider starting with SGD by maintaining (11), and then apply the following Dual Coordinate Ascent rule:

$$\alpha_\ell^{(t)} = \begin{cases} \alpha_i^{(t-1)} - \eta_t(\nabla\phi_i(w^{(t-1)}) + \lambda n\alpha_i^{(t-1)}) & \ell = i \\ \alpha_\ell^{(t-1)} & \ell \neq i \end{cases} \qquad (j = 1, \dots, k) \tag{13}$$

and then update $w$ as $w^{(t)} = w^{(t-1)} + (\alpha_i^{(t)} - \alpha_i^{(t-1)})$.

It can be checked that if we take expectation over random $i \in \{1, \dots, n\}$, then the SGD rule in (12) and the dual coordinate ascent rule (13) both yield the gradient descent rule

$$E[w^{(t}|w^{(t-1)}] = w^{(t-1)} - \eta_t\nabla P(w^{(t-1)}).$$

Therefore both can be regarded as different realizations of the more general stochastic gradient rule in (4). However, the advantage of (13) is that we may take a larger step when $t \to \infty$. This is because according to (10), when the primal-dual parameters $(w, \alpha)$ converge to the optimal parameters $(w_*, \alpha_*)$, we have

$$(\nabla\phi_i(w) + \lambda n\alpha_i) \to 0,$$

which means that even if the learning rate $\eta_t$ stays bounded away from zero, the procedure can converge. This is the same effect as SVRG, in the sense that the variance goes to zero asymptotically: as $w \to w_*$ and $\alpha \to \alpha_*$, we have

$$\frac{1}{n}\sum_{i=1}^{n}(\nabla\phi_i(w) + \lambda n\alpha_i)^2 \to 0.$$

That is, SDCA is also a variance reduction method for SGD, which is similar to SVRG.

From this discussion, we can view SVRG as an explicit variance reduction technique for SGD which is similar to SDCA. However, it is simpler, more intuitive, and easier to analyze. This relationship provides useful insights into the underlying optimization problem that may allow us to make further improvements.

## 5  Experiments

To confirm the theoretical results and insights, we experimented with SVRG (Fig. 1 Option I) in comparison with SGD and SDCA with linear predictors (convex) and neural nets (nonconvex). In all the figures, the $x$-axis is computational cost measured by the number of gradient computations divided by $n$. For SGD, it is the number of passes to go through the training data, and for SVRG in the nonconvex case (neural nets), it includes the additional computation of $\nabla\psi_i(\tilde{w})$ both in each iteration and for computing the gradient average $\tilde{\mu}$. For SVRG in our convex case, however, $\nabla\psi_i(\tilde{w})$ does not have to be re-computed in each iteration. Since in this case the gradient is always a multiple of $x_i$, i.e., $\nabla\psi_i(w) = \phi_i'(w^\top x_i)x_i$ where $\psi_i(w) = \phi_i(w^\top x_i)$, $\nabla\psi_i(\tilde{w})$ can be compactly saved in memory by only saving scalars $\phi_i'(\tilde{w}^\top x_i)$ with the same memory consumption as SDCA and SAG. The interval $m$ was set to $2n$ (convex) and $5n$ (nonconvex). The weights for SVRG were initialized by performing 1 iteration (convex) or 10 iterations (nonconvex) of SGD; therefore, the line for SVRG starts after $x = 1$ (convex) or $x = 10$ (nonconvex) in the respective figures.

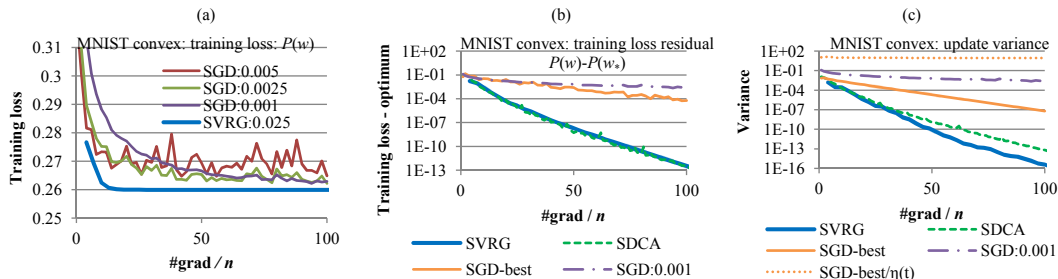

Figure 2: Multiclass logistic regression (convex) on MNIST. (a) Training loss comparison with SGD with fixed learning rates. The numbers in the legends are the learning rate. (b) Training loss residual $P(w) - P(w_*)$; comparison with best-tuned SGD with learning rate scheduling and SDCA. (c) Variance of weight update (including multiplication with the learning rate).

First, we performed L2-regularized multiclass logistic regression (convex optimization) on MNIST[1] with regularization parameter $\lambda =$1e-4. Fig. 2 (a) shows training loss (i.e., the optimization objective $P(w)$) in comparison with SGD with fixed learning rates. The results are indicative of the known weakness of SGD, which also illustrates the strength of SVRG. That is, when a relatively large learning rate $\eta$ is used with SGD, training loss drops fast at first, but it oscillates above the minimum and never goes down to the minimum. With small $\eta$, the minimum may be approached eventually, but it will take many iterations to get there. Therefore, to accelerate SGD, one has to start with relatively large $\eta$ and gradually decrease it (*learning rate scheduling*), as commonly practiced. By contrast, using a single relatively large value of $\eta$, SVRG smoothly goes down faster than SGD. This is in line with our theoretical prediction that one can use a relatively large $\eta$ with SVRG, which leads to faster convergence.

Fig. 2 (b) and (c) compare SVRG with best-tuned SGD with learning rate scheduling and SDCA. 'SGD-best' is the best-tuned SGD, which was chosen by preferring smaller training loss from a large number of parameter combinations for two types of learning scheduling: exponential decay $\eta(t) = \eta_0 a^{\lfloor t/n \rfloor}$ with parameters $\eta_0$ and $a$ to adjust and t-inverse $\eta(t) = \eta_0 (1 + b \lfloor t/n \rfloor)^{-1}$ with $\eta_0$ and $b$ to adjust. (Not surprisingly, the best-tuned SGD with learning rate scheduling outperformed the best-tuned SGD with a fixed learning rate throughout our experiments.) Fig. 2 (b) shows training loss residual, which is training loss minus the optimum (estimated by running gradient descent for a very long time): $P(w) - P(w_*)$. We observe that SVRG's loss residual goes down exponentially, which is in line with Theorem 1, and that SVRG is competitive with SDCA (the two lines are almost overlapping) and decreases faster than SGD-best. In Fig. 2 (c), we show the variance of SVRG update $-\eta(\nabla \psi_i(w) - \nabla \psi_i(\tilde{w}) + \tilde{\mu})$ in comparison with the variance of SGD update $-\eta(t)\nabla \psi_i(w)$ and SDCA. As expected, the variance of both SVRG and SDCA decreases as optimization proceeds, and the variance of SGD with a fixed learning rate ('SGD:0.001') stays high. The variance of the best-tuned SGD decreases, but this is due to the forced exponential decay of the learning rate and the variance of the gradients $\nabla \psi_i(w)$ (the dotted line labeled as 'SGD-best/$\eta(t)$') stays high.

Fig. 3 shows more convex-case results (L2-regularized logistic regression) in terms of training loss residual (top) and test error rate (bottom) on rcv1.binary and covtype.binary from the LIBSVM site[2], protein[3], and CIFAR-10[4]. As protein and covtype do not come with labeled test data, we randomly split the training data into halves to make the training/test split. CIFAR was normalized into $[0, 1]$ by division with 255 (which was also done with MNIST and CIFAR in the other figures), and protein was standardized. $\lambda$ was set to 1e-3 (CIFAR) and 1e-5 (rest). Overall, SVRG is competitive with SDCA and clearly more advantageous than the best-tuned SGD. It is also worth mentioning that a recent study Schmidt et al. [2013] reports that SAG and SDCA are competitive.

To test SVRG with nonconvex objectives, we trained neural nets (with one fully-connected hidden layer of 100 nodes and ten softmax output nodes; sigmoid activation and L2 regularization) with mini-batches of size 10 on MNIST and CIFAR-10, both of which are standard datasets for deep

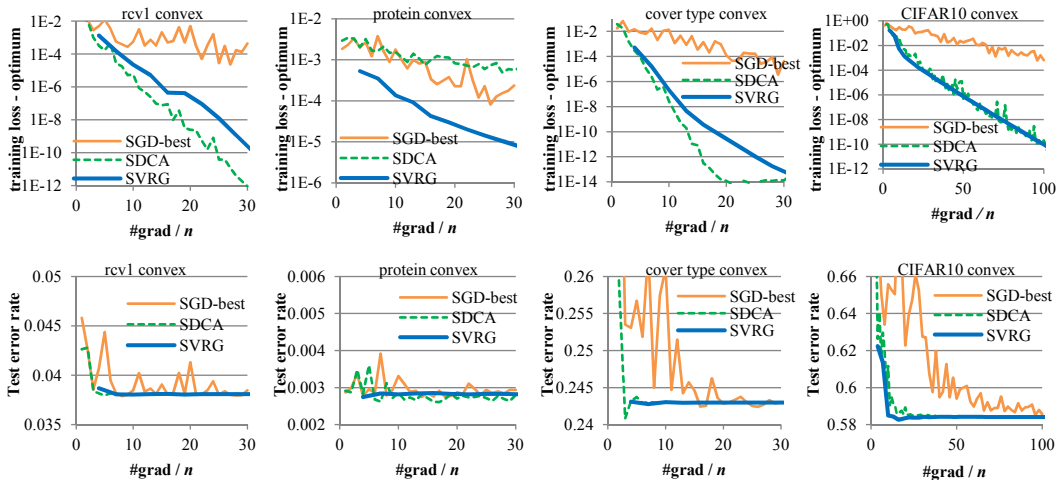

Figure 3: More convex-case results. Loss residual $P(w) - P(w_*)$ (top) and test error rates (down). L2-regularized logistic regression (10-class for CIFAR-10 and binary for the rest).

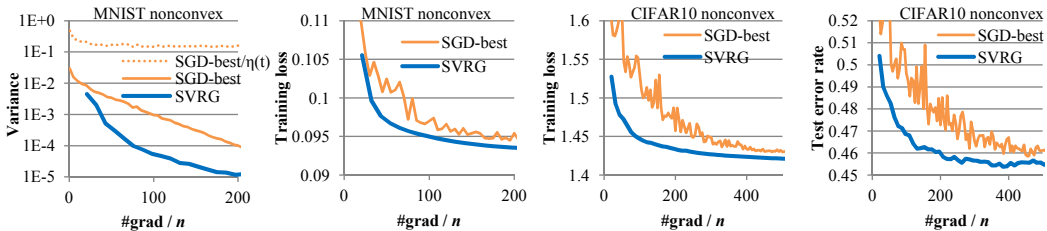

Figure 4: Neural net results (nonconvex).

neural net studies; $\lambda$ was set to 1e-4 and 1e-3, respectively. In Fig. 4 we confirm that the results are similar to the convex case; i.e., SVRG reduces the variance and smoothly converges faster than the best-tuned SGD with learning rate scheduling, which is a de facto standard method for neural net training. As said earlier, methods such as SDCA and SAG are not practical for neural nets due to their memory requirement. We view these results as promising. However, further investigation, in particular with larger/deeper neural nets for which training cost is a critical issue, is still needed.

# 6 Conclusion

This paper introduces an explicit variance reduction method for stochastic gradient descent methods. For smooth and strongly convex functions, we prove that this method enjoys the same fast convergence rate as those of SDCA and SAG. However, our proof is significantly simpler and more intuitive. Moreover, unlike SDCA or SAG, this method does not require the storage of gradients, and thus is more easily applicable to complex problems such as structured prediction or neural network learning.

## Acknowledgment

We thank Leon Bottou and Alekh Agarwal for spotting a mistake in the original theorem.

## Footnotes

[1] http://yann.lecun.com/exdb/mnist/

[2] http://www.csie.ntu.edu.tw/~cjlin/libsvmtools/datasets/

[3] http://osmot.cs.cornell.edu/kddcup/datasets.html

[4] www.cs.toronto.edu/~kriz/cifar.html

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
