[Reviews · NeurIPS 2013]

Submitted by Assigned_Reviewer_5

The authors introduce a new, straightforward algorithm (SVRG) that achieves a linear (geometric) convergence rate for batch stochastic optimization of a smooth, strongly-convex objective on a fixed dataset.  This algorithm has two main advantages over previous algorithms (SDCA and SAG) that achieve the same rates: 1) the analysis is easier, and 2) there is no need to store n past gradients in main memory.

I agree with the authors that the algorithm and the analysis are relatively intuitive compared to SDCA and SAG.  Thus, the theoretical contributions of the paper are certainly its strongest.

For many practical problems, the gradients for particular \psi_i are sparse, that is, typically have less than k nonzeros, with k << d.  As stated SVRG will be very inefficient on such problems, since each update will take O(d) time as opposed to O(k) for SGD, since generally \tilde{\mu} will be dense.  However, with a little extra bookkeeping for each dimension, I believe lazily applying the update scheme should surmount this difficulty.  This is worth mentioning.

I found the experiments unsatisfying.  The obvious algorithms to compare to are SDCA and SAG, but neither are considered.  In fact, only constant-step-size SGD is used, which the authors even point out is inappropriate for strongly convex problems (a learning rate like eta/t should be used, or more generally something like eta/(1 + t)^c.  This means the experimental results section unfortunately does not answer the primary question of interest to me:  is SVRG a state-of-the-art algorithm for machine learning problems?  In order to answer this question, the authors would need to compare to other state-of-the-art algorithms.  This should include at least SAG (which was rigorously compared to other algorithms in its NIPS debut), SDCA, and SGD with an appropriate step size.  Further, here the benchmark of interest is held-out test-set error.  Bonus points for taking into account the fact that the optimal amount of L2 regularization may be different for the different algorithms and for different numbers of passes.
Summary: The paper makes a nice theoretical contribution, probably large enough to warrant acceptance.  However, the lack of solid experiments is disappointing.

Submitted by Assigned_Reviewer_6

The basic idea of adding an extra term to the update term in SGD based on a periodically evaluated full gradient, with the motivation of lowering the variance of SGD is good, and new as far as the literature on SGD that I know of.

In terms of convergence properties the suggested method has similarities with two good methods in recent literature, by Le Roux et al and Shalev-Shwartz and Zhang. But the paper does not do a good job of describing these methods and results associated with them. It would be good to put a small section in the paper for self-containment and better appreciation of the results of this paper.

The theory for the case where strong convexity is not there (Corollary 1) appears weak. The result only talks about closeness with a modified objective function and not min P(w). This needs to be explained.

All the theory is developed for option II in the procedures. But the experiments use option I. Why was option II not used in the experiments? Does it lead to any issues? What are the theoretical complications for proving with option I?

The paper mentions that the main advantage of the new method is in dealing with problems with complex gradients. I understand it, but adding some discussion would be useful.

The experiments only give comparisons against standard SGD. Comparisons against the methods of Le Roux et al and Shalev-Shwartz and Zhang would have been nice to see. For example, how small are the variances of these methods?

Section 4 on SDCA variance reduction is hand waving. The update in (13) is not proper SDCA but only a relative.
Summary: A paper that suggests a good SGD variation.

Submitted by Assigned_Reviewer_7

This work proposes a new algorithm that perturbs a constant-step size stochastic gradient iteration. The perturbation is based on a previous full-gradient evaluation, and allows the method to achieve a linear convergence rate that is faster than the rate obtained by deterministic gradient methods.

The paper proposes a nice approach, that does indeed address the memory problems with using alternate approaches like SDCA and SAG. There is a nice discussion of how all three algorithms essentially force the variance of the estimator to go to zero as the solution is approached.

After equation (6): Normally, strong-convexity is defined with gamma strictly greater than 0. Also, the gradient method has a linear convergence rate with eta < 2/L.

The citation style is wrong, and the bibliography needs to be cleaned up. Note that the SAG and SDCA papers were published in NIPS'12 and ICML'13 conferences, respectively.

Is is probably worth explicitly noting somewhere the requirement that eta < 1/2L in Theorem 1.

The proof is indeed much simpler than the SAG proof. However, in the strongly-convex case I don't think that it is obviously simpler than the SDCA proof. But it is probably worth pointing out that the SDCA method is for a special case of (1).

Because the method requires full passes through the data, it doesn't really seem appropriate to call it a stochastic gradient method. Indeed, on the first pass through the data the method doesn't change the parameters, so it will start out a full pass behind the basic SGD method. A better name might be a hybrid method or a mixed method.

Related to the above, it is worthwhile to add a sentence to the paper contrasting the new approach with the closely-related approaches that use growing sample-sizes, like Friedlander and Schmidt ["Hybrid Deterministic-Stochastic Methods for Data Fitting"] and Byrd et al. ["Sample size selection in optimization methods for machine learning"].

To complement Section 4, it might be worthwhile to try to write the new method as well as SAG as special cases of a more general algorithm. (In SAG, \nabla \phi_{i_t}(\tilde{w}) is replaced by \nabla \phi_{i_t} (w_{i_t}) where w_{i_it} is some previous iteration, and mu is updated at each iteration rather than after every m iterations). It seems like there are numerous variations on this family that would give linear convergence.

The paper should say how eta and m were set in the experiments.

Although the algorithm does indeed achieve a rate similar to SDCA and SAG in terms of number of examples, the rate achieved appears to be slower in general. For example, using the example on page 5 where L/gamma=n, it requires m=8n to have alpha=1/2 with the step size of 1/4L. So the algorithm requires 10 passes through the data on each iteration to achieve this rate (or 9 passes if a memory is used). If you are allowed to do 10 passes through the data of SDCA in this setting, you would get a faster rate of alpha=(1-1/2n)^10n for the same number of points visited as the new algorithm.

(the rate of SDCA is 1 - n\lambda/n(n\lambda+L) = 1-1/2n if lambda=L/n)

The last sentence of Section 3 should be accompanied by a citation or more details. Otherwise, it should be left out.

Regarding Section 4, note that there are other works discussing the relationship between primal gradient methods and dual coordinate-wise methods. For example, see Bach's "Duality between subgradient and conditional gradient methods".

I found the experimental evaluation of the proposed method to be quite weak, only comparing to the SGD method in a regime where 200 or 500 passes through the data set are performed. This does not seem like the regime where SGD would be the method of choice, particularly for the convex objective. Given this setting, and given that the method is a hybrid-type strategy, it would make more sense to also compare to competitive methods (like an L-BFGS or non-linear conjugate gradient code) and other hybrid methods (such as those cited above), since these would be the logical competitors.

For the first experiment, it would also be informative to compare directly to SDCA or SAG, and simply say that their memory requirements do not allow them to be used in the second experiment.

For Figure 3, it would be informative to plot the sub-optimality on a log-scale, to empirically verify that the method is converging linearly.
Summary: The work proposes an interesting algorithm, with similar motivations to the recent SDCA and SAG methods, but without the memory requirements of these methods. I have a few minor issues with the text (easily fixed), and feel that the paper would be much stronger if more work would have been done on the experiments. Nevertheless, the new algorithm may be sufficiently interesting to the NIPS community in its current form as it is very simple and it is the first fast stochastic gradient method that can be applied to less-structured problems like deep networks.
Author Feedback

Author rebuttal: We appreciate all the comments.

The main concern seems to be incomplete experiments: although not in the submission due to space limitation, we do have more comprehensive experiments on a number of datasets, and empirically the new algorithm is state-of-the-art and it is competitive to sdca on convex problems. The new method is better on non convex problems. The theory does carry out to practice, and we think our technique will be of broader interests for all nips audience who care about optimization, and will stimulate future work along this line.

Again, we appreciate the reviewers for critical comments.